# Deep Semi-Supervised Anomaly Detection

**Lukas Ruff**[1]      **Robert A. Vandermeulen**[1*]      **Nico Görnitz**[1 2]
**Alexander Binder**[3]      **Emmanuel Müller**[4]
**Klaus-Robert Müller**[1 5 6]      **Marius Kloft**[7†]

[1]Technical University of Berlin, Germany
[2]123ai.de, Berlin, Germany
[3]Singapore University of Technology & Design, Singapore
[4]Bonn-Aachen International Center for Information Technology, Germany
[5]Korea University, Seoul, Republic of Korea
[6]Max Planck Institute for Informatics, Saarbrücken, Germany
[7]Technical University of Kaiserslautern, Germany
`{lukas.ruff, vandermeulen, nico.goernitz}@tu-berlin.de`
`alexander_binder@sutd.edu.sg    mueller@bit.uni-bonn.de`
`klaus-robert.mueller@tu-berlin.de    kloft@cs.uni-kl.de`

## Abstract

Deep approaches to anomaly detection have recently shown promising results over shallow methods on large and complex datasets. Typically anomaly detection is treated as an unsupervised learning problem. In practice however, one may have—in addition to a large set of unlabeled samples—access to a small pool of labeled samples, e.g. a subset verified by some domain expert as being normal or anomalous. Semi-supervised approaches to anomaly detection aim to utilize such labeled samples, but most proposed methods are limited to merely including labeled normal samples. Only a few methods take advantage of labeled anomalies, with existing deep approaches being domain-specific. In this work we present *Deep SAD*, an end-to-end deep methodology for general semi-supervised anomaly detection. We further introduce an information-theoretic framework for deep anomaly detection based on the idea that the entropy of the latent distribution for normal data should be lower than the entropy of the anomalous distribution, which can serve as a theoretical interpretation for our method. In extensive experiments on MNIST, Fashion-MNIST, and CIFAR-10, along with other anomaly detection benchmark datasets, we demonstrate that our method is on par or outperforms shallow, hybrid, and deep competitors, yielding appreciable performance improvements even when provided with only little labeled data.

## 1    Introduction

Anomaly detection (AD) (Chandola et al., 2009; Pimentel et al., 2014) is the task of identifying unusual samples in data. Typically AD methods attempt to learn a "compact" description of the data in an unsupervised manner assuming that most of the samples are normal (i.e., not anomalous). For example, in one-class classification (Moya et al., 1993; Schölkopf et al., 2001) the objective is to find a set of small measure which contains most of the data and samples not contained in that set are deemed anomalous. Shallow unsupervised AD methods such as the One-Class SVM (Schölkopf et al., 2001; Tax & Duin, 2004), Kernel Density Estimation (Parzen, 1962; Kim & Scott, 2012; Vandermeulen & Scott, 2013), or Isolation Forest (Liu et al., 2008) often require manual feature engineering to be effective on high-dimensional data and are limited in their scalability to large datasets. These limitations have sparked great interest in developing novel *deep* approaches to unsupervised AD (Erfani et al., 2016; Zhai et al., 2016; Chen et al., 2017; Ruff et al., 2018; Deecke et al., 2018; Ruff et al., 2019; Golan & El-Yaniv, 2018; Pang et al., 2019; Hendrycks et al., 2019a;b).

---

*Majority of the work was done while RV was at TU Kaiserslautern, Germany.

†Part of the work was done while MK was a sabbatical visitor of the DASH Center at the University of Southern California, United States.

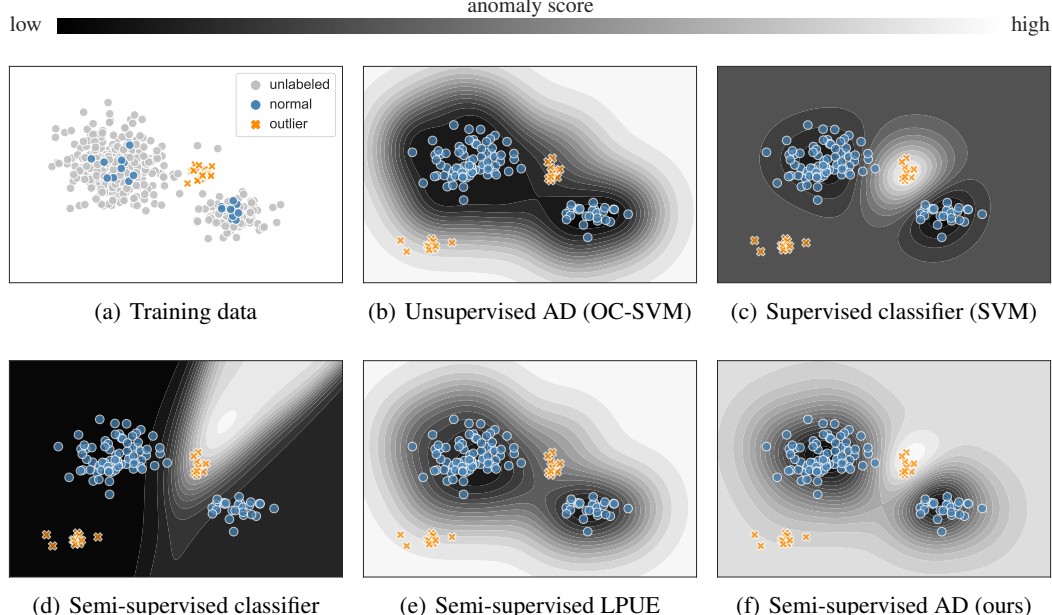

Figure 1: The need for semi-supervised anomaly detection: The training data (shown in (a)) consists of (mostly normal) unlabeled data (gray) as well as a few labeled normal samples (blue) and labeled anomalies (orange). Figures (b)–(f) show the decision boundaries of the various learning paradigms at testing time along with novel anomalies that occur (bottom left in each plot). Our semi-supervised AD approach takes advantage of all training data: unlabeled samples, labeled normal samples, as well as labeled anomalies. This strikes a balance between one-class learning and classification.

Unlike the standard unsupervised AD setting, in many real-world applications one may also have access to some verified (i.e., labeled) normal or anomalous samples in addition to the unlabeled data. Such samples could be hand labeled by a domain expert for instance. This leads to a semi-supervised AD problem: given $n$ (mostly normal but possibly containing some anomalous contamination) unlabeled samples $\boldsymbol{x}_1, \ldots, \boldsymbol{x}_n$ and $m$ labeled samples $(\tilde{\boldsymbol{x}}_1, \tilde{y}_1), \ldots, (\tilde{\boldsymbol{x}}_m, \tilde{y}_m)$, where $\tilde{y} = +1$ and $\tilde{y} = -1$ denote normal and anomalous samples respectively, the task is to learn a model that compactly characterizes the "normal class."

The term *semi-supervised anomaly detection* has been used to describe two different AD settings. Most existing "semi-supervised" AD methods, both shallow (Muñoz-Marí et al., 2010; Blanchard et al., 2010; Chandola et al., 2009) and deep (Song et al., 2017; Akcay et al., 2018; Chalapathy & Chawla, 2019), only incorporate the use of labeled normal samples but not labeled anomalies, i.e. they are more precisely instances of Learning from Positive (i.e., normal) and Unlabeled Examples (LPUE) (Denis, 1998; Zhang & Zuo, 2008). A few works (Wang et al., 2005; Liu & Zheng, 2006; Görnitz et al., 2013) have investigated the general semi-supervised AD setting where one also utilizes labeled anomalies, however existing deep approaches are domain or data-type specific (Ergen et al., 2017; Kiran et al., 2018; Min et al., 2018).

Research on deep semi-supervised learning has almost exclusively focused on classification as the downstream task (Kingma et al., 2014; Rasmus et al., 2015; Odena, 2016; Dai et al., 2017; Oliver et al., 2018). Such semi-supervised classifiers typically assume that similar points are likely to be of the same class, this is known as the *cluster assumption* (Zhu, 2005; Chapelle et al., 2009). This assumption, however, only holds for the "normal class" in AD, but is crucially invalid for the "anomaly class" since anomalies are not necessarily similar to one another. Instead, semi-supervised AD approaches must find a compact description of the normal class while also correctly discriminating the labeled anomalies (Görnitz et al., 2013). Figure 1 illustrates the differences between various learning paradigms applied to AD on a toy example.

We introduce *Deep SAD* (Deep Semi-supervised Anomaly Detection) in this work, an end-to-end deep method for general semi-supervised AD. Our main contributions are the following:

- We introduce Deep SAD, a generalization of the unsupervised Deep SVDD method (Ruff et al., 2018) to the semi-supervised AD setting.

- We present an information-theoretic framework for deep AD, which can serve as an interpretation of our Deep SAD method and similar approaches.

- We conduct extensive experiments in which we establish experimental scenarios for the general semi-supervised AD problem where we also introduce novel baselines.

## 2 AN INFORMATION-THEORETIC VIEW ON DEEP ANOMALY DETECTION

The study of the theoretical foundations of deep learning is an active and ongoing research effort (Montavon et al., 2011; Tishby & Zaslavsky, 2015; Cohen et al., 2016; Eldan & Shamir, 2016; Neyshabur et al., 2017; Raghu et al., 2017; Zhang et al., 2017; Achille & Soatto, 2018; Arora et al., 2018; Belkin et al., 2018; Wiatowski & Bölcskei, 2018; Lapuschkin et al., 2019). One important line of research that has emerged is rooted in information theory (Shannon, 1948). In the supervised classification setting where one has input variable $X$, latent variable $Z$ (e.g., the final layer of a deep network), and output variable $Y$ (i.e., the label), the well-known *Information Bottleneck principle* (Tishby et al., 1999; Tishby & Zaslavsky, 2015; Shwartz-Ziv & Tishby, 2017; Alemi et al., 2017; Saxe et al., 2018) provides an explanation for representation learning as the trade-off between finding a minimal compression $Z$ of the input $X$ while retaining the informativeness of $Z$ for predicting the label $Y$. Put formally, supervised deep learning seeks to minimize the mutual information $\mathcal{I}(X; Z)$ between the input $X$ and the latent representation $Z$ while maximizing the mutual information $\mathcal{I}(Z; Y)$ between $Z$ and the classification task $Y$, i.e.

$$\min_{p(z|x)} \quad \mathcal{I}(X; Z) - \alpha \, \mathcal{I}(Z; Y), \tag{1}$$

where $p(z|x)$ is modeled by a deep network and the hyperparameter $\alpha > 0$ controls the trade-off between compression (i.e., complexity) and classification accuracy.

For unsupervised deep learning, due to the absence of labels $Y$ and thus the lack of a clear task, other information-theoretic learning principles have been formulated. Of these, the *Infomax principle* (Linsker, 1988; Bell & Sejnowski, 1995; Hjelm et al., 2019) is one of the most prevalent and widely used principles. In contrast to (1), the objective of Infomax is to *maximize* the mutual information $\mathcal{I}(X; Z)$ between the data $X$ and its latent representation $Z$:

$$\max_{p(z|x)} \quad \mathcal{I}(X; Z) + \beta \, \mathcal{R}(Z). \tag{2}$$

This is typically done under some additional constraint or regularization $\mathcal{R}(Z)$ on the representation $Z$ with hyperparameter $\beta > 0$ to obtain statistical properties desired for some specific downstream task. Examples where the Infomax principle has been applied include tasks such as independent component analysis (Bell & Sejnowski, 1995), clustering (Slonim et al., 2005; Ji et al., 2018), generative modeling (Chen et al., 2016; Hoffman & Johnson, 2016; Zhao et al., 2017; Alemi et al., 2018), and unsupervised representation learning in general (Hjelm et al., 2019).

We observe that the Infomax principle has also been applied in previous deep representations for AD. Most notably autoencoders (Rumelhart et al., 1986; Hinton & Salakhutdinov, 2006), which are the predominant approach to deep AD (Hawkins et al., 2002; Sakurada & Yairi, 2014; Andrews et al., 2016; Erfani et al., 2016; Zhai et al., 2016; Chen et al., 2017; Chalapathy & Chawla, 2019), can be understood as implicitly maximizing the mutual information $\mathcal{I}(X; Z)$ via the reconstruction objective (Vincent et al., 2008) under some regularization of the latent code $Z$. Choices for regularization include sparsity (Makhzani & Frey, 2014), the distance to some latent prior distribution, e.g. measured via the KL divergence (Kingma & Welling, 2013; Rezende et al., 2014), an adversarial loss (Makhzani et al., 2015), or simply a bottleneck in dimensionality. Such restrictions for AD share the idea that the latent representation of the normal data should be in some sense "compact."

As illustrated in Figure 1, a supervised (or semi-supervised) classification approach to AD only learns to recognize anomalies similar to those seen during training, due to the class cluster assumption (Chapelle et al., 2009). However, *anything* not normal is by definition an anomaly and thus anomalies do not have to be similar. This makes supervised (or semi-supervised) classification learning principles such as (1) ill-defined for AD. We instead build upon principle (2) to motivate a deep method for general semi-supervised AD, where we include the label information $Y$ through a novel representation learning regularization objective $\mathcal{R}(Z) = \mathcal{R}(Z; Y)$ that is based on entropy.

## 3 DEEP SEMI-SUPERVISED ANOMALY DETECTION

In the following, we introduce *Deep SAD*, a deep method for general semi-supervised AD. To formulate our objective, we first briefly explain the unsupervised Deep SVDD method (Ruff et al., 2018) which we then generalize to the semi-supervised AD setting.

### 3.1 UNSUPERVISED DEEP SVDD AND ENTROPY MINIMIZATION

For input space $\mathcal{X} \subseteq \mathbb{R}^D$ and output space $\mathcal{Z} \subseteq \mathbb{R}^d$, let $\phi(\,\cdot\,;\mathcal{W}) : \mathcal{X} \to \mathcal{Z}$ be a neural network with $L$ hidden layers and corresponding set of weights $\mathcal{W} = \{\boldsymbol{W}^1, \ldots, \boldsymbol{W}^L\}$. The objective of Deep SVDD is to train the neural network $\phi$ to learn a transformation that minimizes the volume of a data-enclosing hypersphere in output space $\mathcal{Z}$ centered on a predetermined point $\boldsymbol{c}$. Given $n$ (unlabeled) training samples $\boldsymbol{x}_1, \ldots, \boldsymbol{x}_n \in \mathcal{X}$, the *One-Class Deep SVDD* objective is

$$\min_{\mathcal{W}} \quad \frac{1}{n} \sum_{i=1}^n \|\phi(\boldsymbol{x}_i; \mathcal{W}) - \boldsymbol{c}\|^2 + \frac{\lambda}{2} \sum_{\ell=1}^L \|\boldsymbol{W}^\ell\|_F^2, \quad \lambda > 0. \tag{3}$$

Penalizing the mean squared distance of the mapped samples to the hypersphere center $\boldsymbol{c}$ forces the network to extract those common factors of variation which are most stable within the dataset. As a consequence normal data points tend to get mapped near the hypersphere center, whereas anomalies are mapped further away (Ruff et al., 2018). The second term is a standard weight decay regularizer.

Deep SVDD is optimized via SGD using backpropagation. For initialization, Ruff et al. (2018) first pre-train an autoencoder and then initialize the weights $\mathcal{W}$ of the network $\phi$ with the converged weights of the encoder. After initialization, the hypersphere center $\boldsymbol{c}$ is set as the mean of the network outputs obtained from an initial forward pass of the data. Once the network is trained, the anomaly score for a test point $\boldsymbol{x}$ is given by the distance from $\phi(\boldsymbol{x}; \mathcal{W})$ to the center of the hypersphere:

$$s(\boldsymbol{x}) = \|\phi(\boldsymbol{x}; \mathcal{W}) - \boldsymbol{c}\|. \tag{4}$$

We now argue that Deep SVDD may not only be interpreted in geometric terms as minimum volume estimation (Scott & Nowak, 2006), but also in probabilistic terms as entropy minimization over the latent distribution. For a latent random variable $Z$ with covariance $\Sigma$, pdf $p(\boldsymbol{z})$, and support $\mathcal{Z} \subseteq \mathbb{R}^d$, we have the following bound on entropy

$$\mathcal{H}(Z) = \mathbb{E}[-\log p(Z)] = -\int_{\mathcal{Z}} p(\boldsymbol{z}) \log p(\boldsymbol{z}) \, \mathrm{d}\boldsymbol{z} \leq \frac{1}{2} \log((2\pi e)^d \det \Sigma), \tag{5}$$

which holds with equality iff $Z$ is jointly Gaussian (Cover & Thomas, 2012). Assuming the latent distribution $Z$ follows an isotropic Gaussian, $Z \sim N(\boldsymbol{\mu}, \sigma^2 I)$ with $\sigma > 0$, we get

$$\mathcal{H}(Z) = \frac{1}{2} \log((2\pi e)^d \det \sigma^2 I) = \frac{1}{2} \log((2\pi e \sigma^2)^d \cdot 1) = \frac{d}{2}(1 + \log(2\pi \sigma^2)) \propto \log \sigma^2, \tag{6}$$

i.e. for a fixed dimensionality $d$, the entropy of $Z$ is proportional to its log-variance.

Now observe that the Deep SVDD objective (3) (disregarding weight decay regularization) is equivalent to minimizing the empirical variance and thus minimizes an upper bound on the entropy of a latent Gaussian. Since the Deep SVDD network is pre-trained on an autoencoding objective that implicitly maximizes the mutual information $\mathcal{I}(X; Z)$ (Vincent et al., 2008), we may interpret Deep SVDD as following the Infomax principle (2) with the additional "compactness" objective that the latent distribution should have minimal entropy.

### 3.2 DEEP SAD

We now introduce our method for deep semi-supervised anomaly detection: *Deep SAD*. Assume that, in addition to the $n$ unlabeled samples $\boldsymbol{x}_1, \ldots, \boldsymbol{x}_n \in \mathcal{X}$ with $\mathcal{X} \subseteq \mathbb{R}^D$, we also have access to $m$ labeled samples $(\tilde{\boldsymbol{x}}_1, \tilde{y}_1), \ldots, (\tilde{\boldsymbol{x}}_m, \tilde{y}_m) \in \mathcal{X} \times \mathcal{Y}$ with $\mathcal{Y} = \{-1, +1\}$ where $\tilde{y} = +1$ denotes known normal samples and $\tilde{y} = -1$ known anomalies. We define our *Deep SAD* objective as follows:

$$\min_{\mathcal{W}} \quad \frac{1}{n+m} \sum_{i=1}^n \|\phi(\boldsymbol{x}_i; \mathcal{W}) - \boldsymbol{c}\|^2 + \frac{\eta}{n+m} \sum_{j=1}^m \left(\|\phi(\tilde{\boldsymbol{x}}_j; \mathcal{W}) - \boldsymbol{c}\|^2\right)^{\tilde{y}_j} + \frac{\lambda}{2} \sum_{\ell=1}^L \|\boldsymbol{W}^\ell\|_F^2. \tag{7}$$

We employ the same loss term as Deep SVDD for the unlabeled data in our Deep SAD objective and thus recover Deep SVDD (3) as the special case when there is no labeled training data available ($m = 0$). In doing this we also incorporate the assumption that most of the unlabeled data is normal.

For the labeled data, we introduce a new loss term that is weighted via the hyperparameter $\eta > 0$ which controls the balance between the labeled and the unlabeled term. Setting $\eta > 1$ puts more emphasis on the labeled data whereas $\eta < 1$ emphasizes the unlabeled data. For the labeled normal samples ($\tilde{y} = +1$), we also impose a quadratic loss on the distances of the mapped points to the center $c$, thus intending to overall learn a latent distribution which concentrates the normal data. Again, one might consider $\eta > 1$ to emphasize labeled normal over unlabeled samples. For the labeled anomalies ($\tilde{y} = -1$) in contrast, we penalize the *inverse* of the distances such that anomalies must be mapped further away from the center.[1] Note that this is in line with the common assumption that anomalies are not concentrated (Schölkopf & Smola, 2002; Steinwart et al., 2005). In our experiments we found that simply setting $\eta = 1$ yields a consistent and substantial performance improvement. A sensitivity analysis on $\eta$ is in Section 4.3.

We define the Deep SAD anomaly score again by the distance of the mapped point to the center $c$ as given in Eq. (4) and optimize our Deep SAD objective (7) via SGD using backpropagation. We provide a summary of the Deep SAD optimization procedure and further details in Appendix C.

In addition to the inverse squared norm loss we experimented with several other losses including the negative squared norm loss, negative robust losses, and the hinge loss. The negative squared norm loss, which is unbounded from below, resulted in an ill-posed optimization problem and caused optimization to diverge. Negative robust losses, such as the Hampel loss, introduce one or more scale parameters which are difficult to select or optimize in conjunction with the changing representation learned by the network. Like Ruff et al. (2018), we observed that the hinge loss was difficult to optimize and resulted in poorer performance. The inverse squared norm loss instead is bounded from below and smooth, which are crucial properties for losses used in deep learning (Goodfellow et al., 2016), and ultimately performed the best while remaining conceptually simple.

Following our insights on the connection between Deep SVDD and entropy minimization from Section 3.1, we may interpret our Deep SAD objective as modeling the latent distribution of normal data, $Z^+ = Z|\{Y=+1\}$, to have *low entropy*, and the latent distribution of anomalies, $Z^- = Z|\{Y=-1\}$, to have *high entropy*. Minimizing the distances to the center $c$ (i.e., minimizing the empirical variance) for the mapped points of labeled normal samples ($\tilde{y} = +1$) induces a latent distribution with low entropy for the normal data. In contrast, penalizing low variance via the inverse squared norm loss for the mapped points of labeled anomalies ($\tilde{y} = -1$) induces a latent distribution with high entropy for the anomalous data. That is, the network must attempt to map known anomalies to some heavy-tailed distribution. We argue that such a model better captures the nature of anomalies, which can be thought of as being generated from an infinite mixture of distributions that are different from the normal data distribution, indubitably a distribution that has high entropy. Our objective notably does not impose any cluster assumption on the anomaly-generating distribution $X|\{Y=-1\}$ as is typically made in supervised or semi-supervised classification approaches (Zhu, 2005; Chapelle et al., 2009). We can express this interpretation in terms of principle (2) with an entropy regularization objective on the latent distribution:

$$\max_{p(z|x)} \quad \mathcal{I}(X; Z) + \beta \left( \mathcal{H}(Z^-) - \mathcal{H}(Z^+) \right). \tag{8}$$

To maximize the mutual information $\mathcal{I}(X; Z)$, Deep SAD also relies on autoencoder pre-training (Vincent et al., 2008; Ruff et al., 2018).

## 4 EXPERIMENTS

We evaluate Deep SAD on MNIST, Fashion-MNIST, and CIFAR-10 as well as on classic AD benchmark datasets. We compare to shallow, hybrid, as well as deep unsupervised, semi-supervised and supervised competitors. We refer to other recent works (Ruff et al., 2018; Golan & El-Yaniv, 2018; Hendrycks et al., 2019a) for further comparisons between unsupervised deep AD methods.[2]

---

[1]To ensure numerical stability, we add a machine epsilon (eps $\sim 10^{-6}$) to the denominator of the inverse.
[2]Our code is available at: https://github.com/lukasruff/Deep-SAD-PyTorch

## 4.1 Competing Methods

We consider the OC-SVM (Schölkopf et al., 2001) and SVDD (Tax & Duin, 2004) with Gaussian kernel (which in this case are equivalent), Isolation Forest (Liu et al., 2008), and KDE (Parzen, 1962) for shallow unsupervised baselines. For deep unsupervised competitors, we consider well-established (convolutional) autoencoders and the state-of-the-art unsupervised Deep SVDD method (Ruff et al., 2018). To avoid confusion, we note again that some literature (Song et al., 2017; Chalapathy & Chawla, 2019) refer to the methods above as being "semi-supervised" if they are trained on only labeled normal samples. For general semi-supervised AD approaches that also take advantage of labeled anomalies, we consider the state-of-the-art shallow SSAD method (Görnitz et al., 2013) with Gaussian kernel. As mentioned earlier, there are no deep competitors for general semi-supervised AD that are applicable to general data types. To get a comprehensive comparison we therefore introduce a novel *hybrid SSAD* baseline that applies SSAD to the latent codes of autoencoder models. Such hybrid methods have demonstrated solid performance improvements over their raw feature counterparts on high-dimensional data (Erfani et al., 2016; Nicolau et al., 2016). We also include such hybrid variants for all unsupervised shallow competitors. To also compare to a deep semi-supervised learning method that targets classification as the downstream task, we add the well-known Semi-Supervised Deep Generative Model (SS-DGM) (Kingma et al., 2014) where we use the latent class probability estimate (normal vs. anomalous) as the anomaly score. To complete the full learning spectrum, we also include a fully supervised deep classifier trained on the binary cross-entropy loss.

In our experiments we deliberately grant the shallow and hybrid methods an unfair advantage by selecting their hyperparameters to maximize AUC on a subset (10%) of the test set to minimize hyperparameter selection issues. To control for architectural effects between the deep methods, we always use the same (LeNet-type) deep networks. Full details on network architectures and hyperparameter selection can be found in Appendices D and E. Due to space constraints, in the main text we only report results for methods which showed competitive performance and defer results for the underperforming methods in Appendix F.

## 4.2 Experimental Scenarios on MNIST, Fashion-MNIST, and CIFAR-10

**Semi-supervised anomaly detection setup** MNIST, Fashion-MNIST, and CIFAR-10 all have ten classes from which we derive ten AD setups on each dataset following previous works (Ruff et al., 2018; Chalapathy et al., 2018; Golan & El-Yaniv, 2018). In every setup, we set one of the ten classes to be the normal class and let the remaining nine classes represent anomalies. We use the original training data of the respective normal class as the unlabeled part of our training set. Thus we start with a clean AD setting that fulfills the assumption that most (in this case all) unlabeled samples are normal. The training data of the respective nine anomaly classes then forms the data pool from which we draw anomalies for training to create different scenarios. We compute the commonly used AUC measure on the original respective test sets using ground truth labels to make a quantitative comparison, i.e. $\tilde{y} = +1$ for the normal class and $\tilde{y} = -1$ for the respective nine anomaly classes. We rescale pixels to $[0, 1]$ via min-max feature scaling as the only data pre-processing step.

**Experimental scenarios** We examine three scenarios in which we vary the following three experimental parameters: (i) the ratio of labeled training data $\gamma_l$, (ii) the ratio of pollution $\gamma_p$ in the unlabeled training data with (unknown) anomalies, and (iii) the number of anomaly classes $k_l$ included in the labeled training data.

**(i) Adding labeled anomalies** In this scenario, we investigate the effect that including labeled anomalies during training has on detection performance to see the benefit of a general semi-supervised AD approach over other paradigms. To do this we increase the ratio of labeled training data $\gamma_l = m/(n+m)$ by adding more and more known anomalies $\tilde{x}_1, \ldots, \tilde{x}_m$ with $\tilde{y}_j = -1$ to the training set. The labeled anomalies are sampled from one of the nine anomaly classes ($k_l = 1$). For testing, we then consider all nine remaining classes as anomalies, i.e. there are eight novel classes at testing time. We do this to simulate the unpredictable nature of anomalies. For the unlabeled part of the training set, we keep the training data of the respective normal class, which we leave unpolluted in this experimental setup, i.e. $\gamma_p = 0$. We iterate this training set generation process per AD setup always over all the nine respective anomaly classes and report the average results over the ten AD setups $\times$ nine anomaly classes, i.e. over 90 experiments per labeled ratio $\gamma_l$.

**(ii) Polluted training data**   Here we investigate the robustness of the different methods to an increasing pollution ratio $\gamma_p$ of the training set with unlabeled anomalies. To do so we pollute the unlabeled part of the training set with anomalies drawn from all nine respective anomaly classes in each AD setup. We fix the ratio of labeled training samples at $\gamma_l = 0.05$ where we again draw samples only from $k_l = 1$ anomaly class in this scenario. We repeat this training set generation process per AD setup over all the nine respective anomaly classes and report the average results over the resulting 90 experiments per pollution ratio $\gamma_p$. We hypothesize that learning from labeled anomalies in a semi-supervised AD approach alleviates the negative impact pollution has on detection performance since similar unknown anomalies in the unlabeled data might be detected.

**(iii) Number of known anomaly classes**   In the last scenario, we compare the detection performance at various numbers of known anomaly classes. In scenarios (i) and (ii), we always sample labeled anomalies only from one out of the nine anomaly classes ($k_l = 1$). In this scenario, we now increase the number of anomaly classes $k_l$ included in the labeled part of the training set. Since we have a limited number of anomaly classes (nine) in each AD setup, we expect the supervised classifier to catch up at some point. We fix the overall ratio of labeled training examples again at $\gamma_l = 0.05$ and consider a pollution ratio of $\gamma_p = 0.1$ for the unlabeled training data in this scenario. We repeat this training set generation process for ten seeds in each of the ten AD setups and report the average results over the resulting 100 experiments per number $k_l$. For each seed, the $k_l$ classes are drawn uniformly at random from the nine respective anomaly classes.

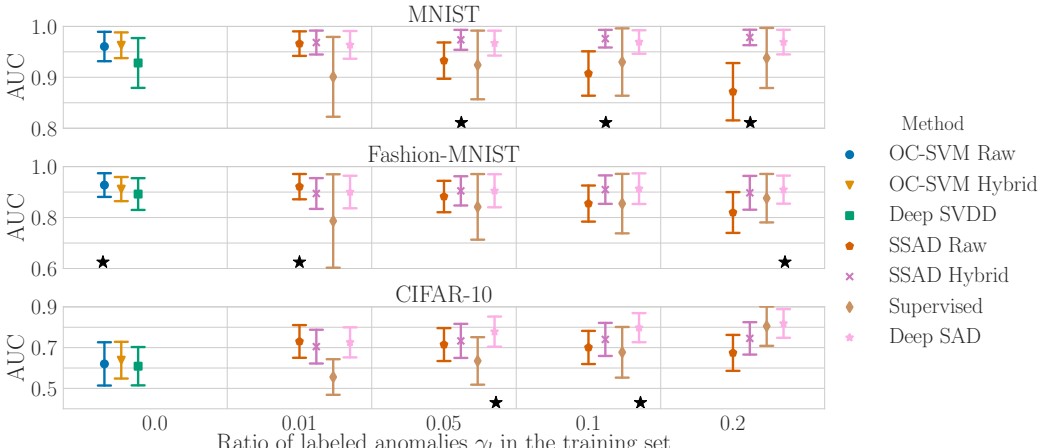

Figure 2: Results of scenario (i), where we increase the ratio of labeled anomalies $\gamma_l$ in the training set. We report avg. AUC with st. dev. over 90 experiments at various ratios $\gamma_l$. A "$\star$" indicates a statistically significant ($\alpha = 0.05$) difference between the 1st and 2nd best method.

**Results**   The results of scenarios (i)–(iii) are shown in Figures 2–4. In addition to the avg. AUC with st. dev., we report the outcome of Wilcoxon signed-rank tests (Wilcoxon, 1945) applied to the first and second best performing method to indicate statistically significant ($\alpha = 0.05$) differences in performance. Figure 2 demonstrates the benefit of our semi-supervised approach to AD especially on the most complex CIFAR-10 dataset, where Deep SAD performs best. Figure 2 moreover confirms that a supervised classification approach is vulnerable to novel anomalies at testing time when only little labeled training data is available. In comparison, Deep SAD generalizes to novel anomalies while also taking advantage of the labeled examples. Note that our novel hybrid SSAD baseline also performs well. Figure 3 shows that the detection performance of all methods decreases with increasing data pollution. Deep SAD proves to be most robust again especially on CIFAR-10. Finally, Figure 4 shows that the more diverse the labeled anomalies in the training set, the better the detection performance becomes. We can again see that the supervised method is very sensitive to the number of anomaly classes but catches up at some point as suspected. This does not occur with CIFAR-10, however, where $\gamma_l = 0.05$ labeled training samples seems to be insufficient for classification. Overall, we see that Deep SAD is particularly beneficial on the more complex data.

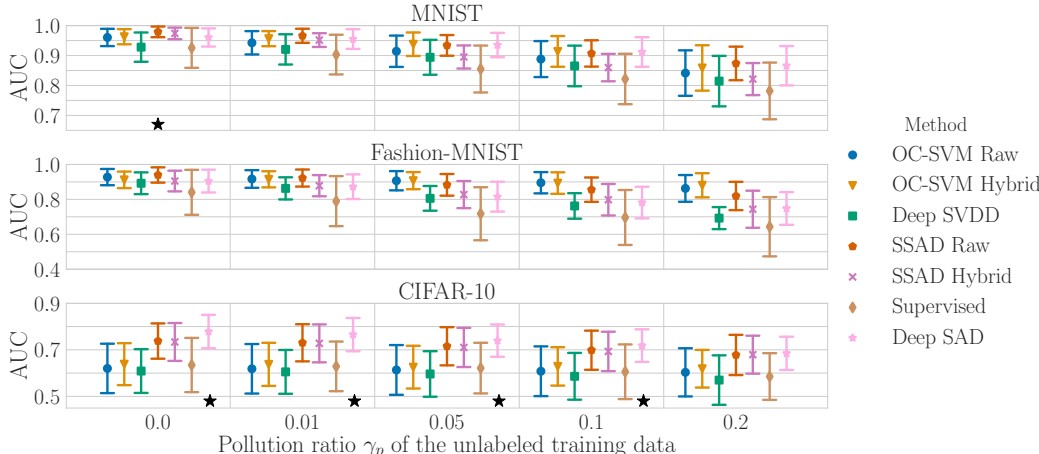

Figure 3: Results of scenario (ii), where we pollute the unlabeled part of the training set with (un-known) anomalies. We report avg. AUC with st. dev. over 90 experiments at various ratios $\gamma_p$. A "$\star$" indicates a statistically significant ($\alpha = 0.05$) difference between the 1st and 2nd best method.

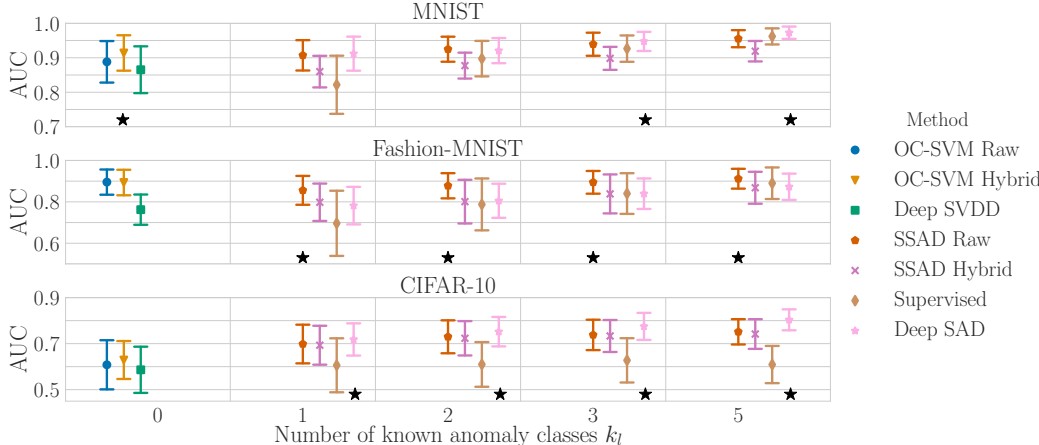

Figure 4: Results of scenario (iii), where we increase the number of anomaly classes $k_l$ included in the labeled training data. We report avg. AUC with st. dev. over 100 experiments for various $k_l$. A "$\star$" indicates a statistically significant ($\alpha = 0.05$) difference between the 1st and 2nd best method.

## 4.3 SENSITIVITY ANALYSIS

We run Deep SAD experiments on the ten AD setups described above on each dataset for $\eta \in \{10^{-2}, \ldots, 10^{2}\}$ to analyze the sensitivity of Deep SAD with respect to the hyperparameter $\eta > 0$. In this analysis, we set the experimental parameters to their default, $\gamma_l = 0.05$, $\gamma_p = 0.1$, and $k_l = 1$, and again iterate over all nine anomaly classes in every AD setup. The results shown in Figure 5 suggest that Deep SAD is fairly robust against changes of the hyperparameter $\eta$.

In addition, we run experiments under the same experimental settings while varying the dimension $d \in \{2^4, \ldots, 2^9\}$ of the output space $\mathcal{Z} \subseteq \mathbb{R}^d$ to infer the sensitivity of Deep SAD with respect to the representation dimensionality, where we keep $\eta = 1$. The results are given in Figure 6 in Appendix A. There we also com-

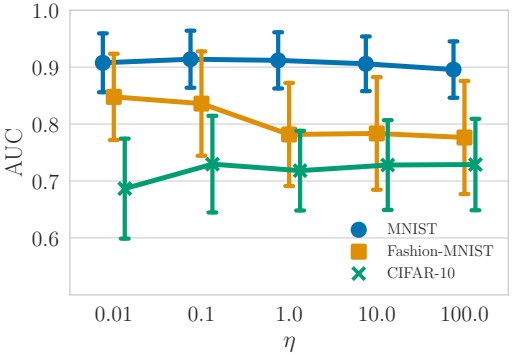

Figure 5: Deep SAD sensitivity analysis w.r.t. $\eta$. We report avg. AUC with st. dev. over 90 experiments for various values of hyperparameter $\eta$.

pare to our hybrid SSAD baseline, which was the strongest competitor. Interestingly we observe that detection performance increases with dimension $d$, converging to an upper bound in performance. This suggests that one would want to set $d$ large enough to have sufficiently high mutual information $\mathcal{I}(X; Z)$ before compressing to a compact characterization.

### 4.4 Classic Anomaly Detection Benchmark Datasets

In a final experiment, we also examine the detection performance of the various methods on some well-established AD benchmark datasets (Rayana, 2016). We run these experiments to evaluate the deep versus the shallow approaches on non-image datasets that are rarely considered in deep AD literature. Here we observe that the shallow kernel methods seem to have a slight edge on the relatively small, low-dimensional benchmarks. Nonetheless, Deep SAD proves competitive and the small differences observed might be explained by the advantage we grant the shallow methods in their hyperparameter selection. We give the full details and results in Appendix B.

Our results and other recent works (Ruff et al., 2018; Golan & El-Yaniv, 2018; Hendrycks et al., 2019a) overall demonstrate that deep methods are especially superior on complex data with hierarchical structure. Unlike other deep approaches (Ergen et al., 2017; Kiran et al., 2018; Min et al., 2018; Deecke et al., 2018; Golan & El-Yaniv, 2018), however, our Deep SAD method is not domain or data-type specific. Due to its good performance using both deep and shallow networks we expect Deep SAD to extend well to other data types.

## 5 Conclusion and Future Work

In this work we introduced Deep SAD, a deep method for general semi-supervised anomaly detection. Our method is a generalization of the unsupervised Deep SVDD method (Ruff et al., 2018) to the semi-supervised setting. The results of our experimental evaluation suggest that general semi-supervised anomaly detection should always be preferred whenever some labeled information on both normal samples or anomalies is available.

Moreover, we formulated an information-theoretic framework for deep anomaly detection based on the Infomax principle. Using this framework, we interpreted our method as minimizing the entropy of the latent distribution for normal data and maximizing the entropy of the latent distribution for anomalous data. We introduced this framework with the aim of forming a basis for new methods as well as rigorous theoretical analyses in the future, e.g. studying deep anomaly detection under the rate-distortion curve (Alemi et al., 2018).

### Acknowledgments

LR acknowledges support by the German Ministry of Education and Research (BMBF) in the project ALICE III (01IS18049B). MK and RV acknowledge support by the German Research Foundation (DFG) award KL 2698/2-1 and by the German Ministry of Education and Research (BMBF) awards 031L0023A, 01IS18051A, and 031B0770E. AB is grateful for support by the National Research Foundation of Singapore, STEE-SUTD Cyber Security Laboratory, and the Ministry of Education, Singapore, under its program MOE2016-T2-2-154. NG acknowledges support by the German Ministry of Education and Research (BMBF) through the Berlin Center for Machine Learning (01IS18037I). KRM acknowledges partial financial support by the German Ministry of Education and Research (BMBF) under grants 01IS14013A-E, 01IS18025A, 01IS18037A, 01GQ1115 and 01GQ0850; Deutsche Forschungsgesellschaft (DFG) under grant Math+, EXC 2046/1, project-ID 390685689, and by the Technology Promotion (IITP) grant funded by the Korea government (No. 2017-0-00451, No. 2017-0-01779).

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

# A  ADDITIONAL RESULTS ON MNIST, FASHION-MNIST, AND CIFAR-10

## A.1  SENSITIVITY ANALYSIS W.R.T REPRESENTATION DIMENSIONALITY

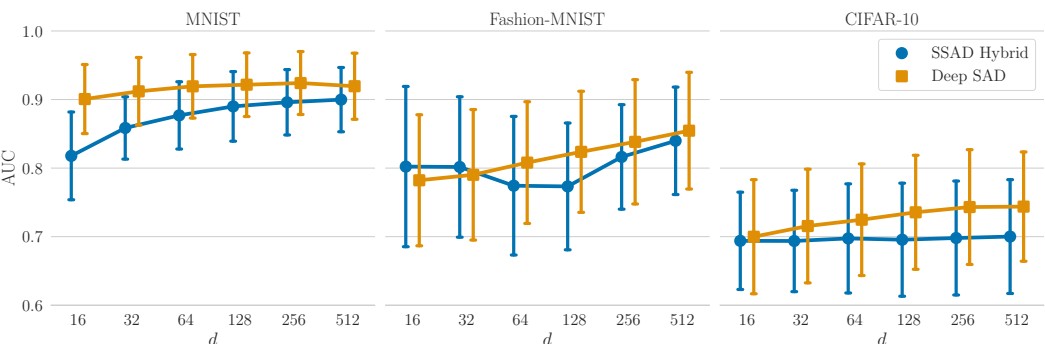

Figure 6: Sensitivity analysis w.r.t. the network representation dimensionality $d$ for our Deep SAD method and the closest competitor hybrid SSAD. We report avg. AUC with st. dev. over 90 experiments for various values of $d$.

## A.2  AUC SCATTERPLOTS OF BEST VS. SECOND BEST METHODS ON CIFAR-10

We provide AUC scatterplots in Figures 7–9 of the best (1st) vs. second best (2nd) performing methods in the experimental scenarios (i)–(iii) on the most complex CIFAR-10 dataset. If most points fall above the identity line, this is a very strong indication that the best method indeed significantly outperforms the second best, which often is the case for our Deep SAD method.

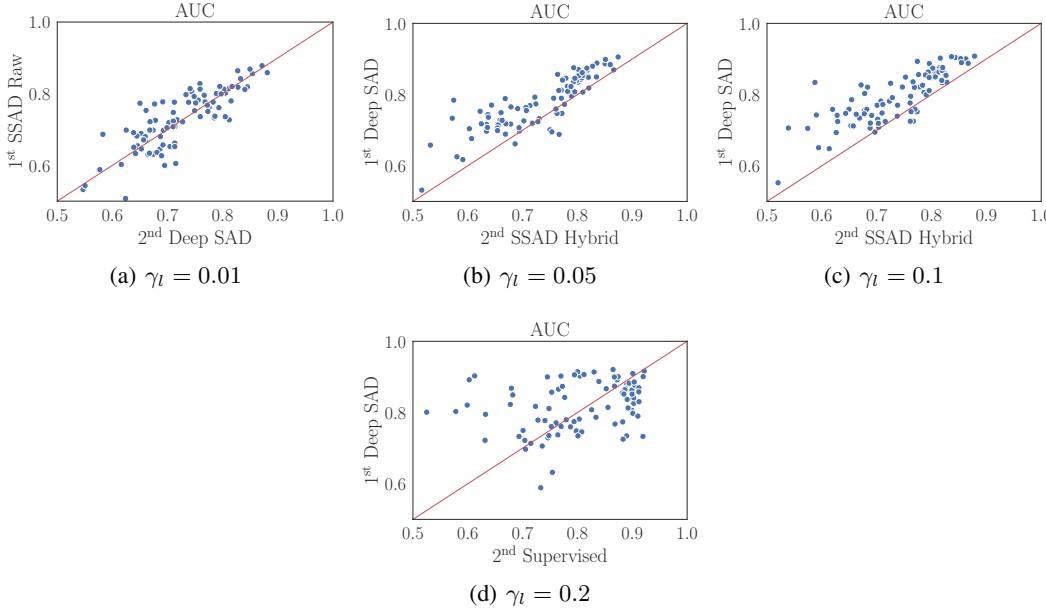

Figure 7: AUC scatterplots of best (1st) vs. second best (2nd) performing methods in experimental scenario (i) on CIFAR-10, where we increase the ratio of labeled anomalies $\gamma_l$ in the training set.

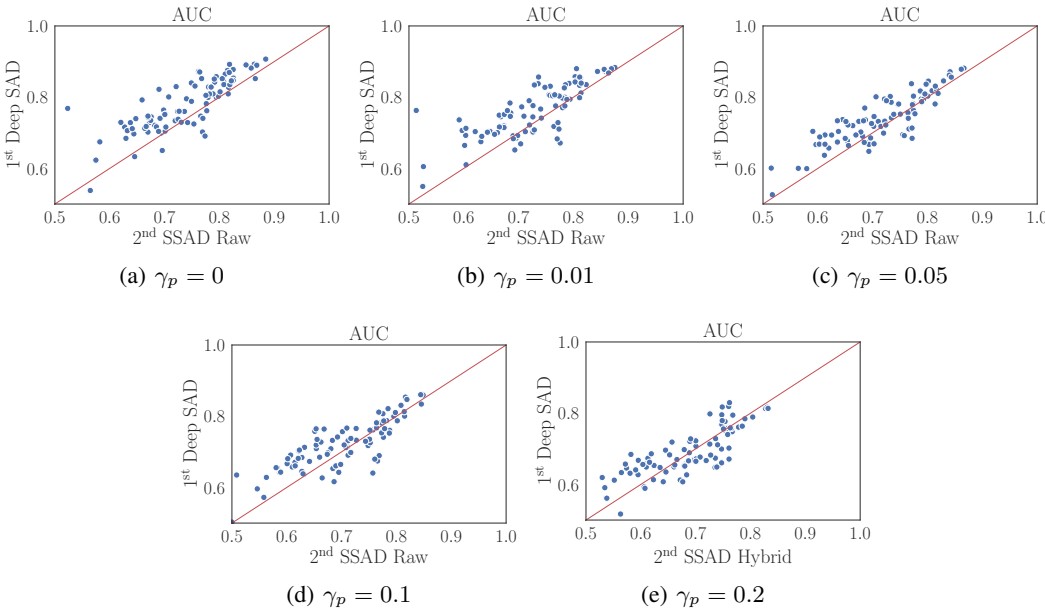

Figure 8: AUC scatterplots of best (1$^\text{st}$) vs. second best (2$^\text{nd}$) performing methods in experimental scenario (ii) on CIFAR-10, where we pollute the unlabeled part of the training set with (unknown) anomalies at various ratios $\gamma_p$.

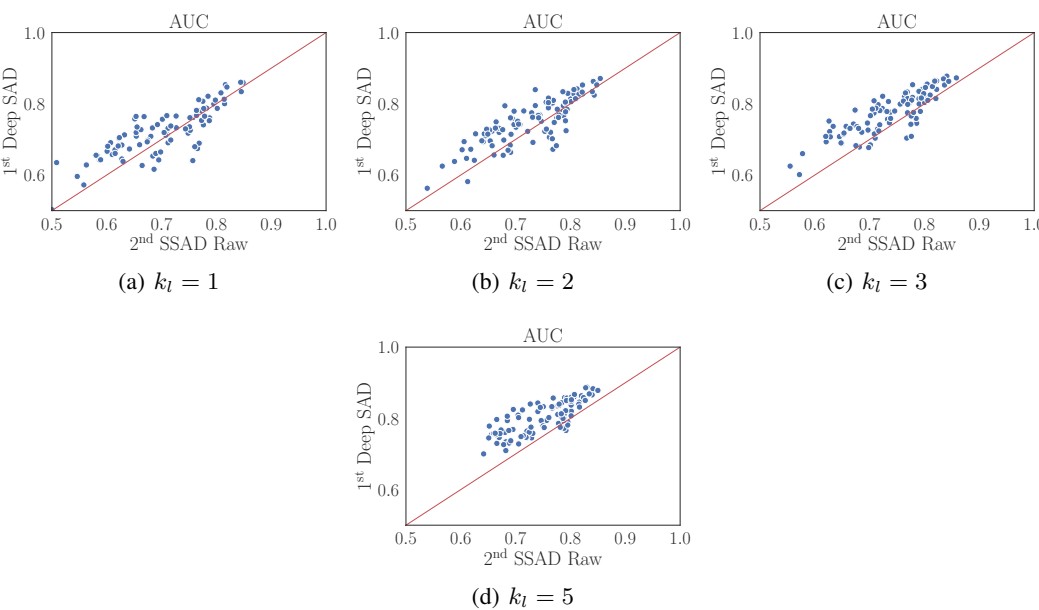

Figure 9: AUC scatterplots of best (1$^\text{st}$) vs. second best (2$^\text{nd}$) performing methods in experimental scenario (iii) on CIFAR-10, where we increase the number of anomaly classes $k_l$ included in the labeled training data.

# B    RESULTS ON CLASSIC ANOMALY DETECTION BENCHMARK DATASETS

In this experiment, we examine the detection performance on some well-established AD benchmark datasets (Rayana, 2016) listed in Table 1. We do this to evaluate the deep against the shallow approaches also on non-image, tabular datasets that are rarely considered in the deep AD literature. For the evaluation, we consider random train-to-test set splits of 60:40 while maintaining the original proportion of anomalies in each set. We then run experiments for 10 seeds with $\gamma_l = 0.01$ and $\gamma_p = 0$, i.e. 1% of the training set are labeled anomalies and the unlabeled training data is unpolluted. Since there are no specific different anomaly classes in these datasets, we have $k_l = 1$. We standardize features to have zero mean and unit variance as the only pre-processing step.

Table 1: Anomaly detection benchmarks.

| Dataset | $N$ | $D$ | #outliers (%) |
|---|---|---|---|
| arrhythmia | 452 | 274 | 66 (14.6%) |
| cardio | 1,831 | 21 | 176 (9.6%) |
| satellite | 6,435 | 36 | 2,036 (31.6%) |
| satimage-2 | 5,803 | 36 | 71 (1.2%) |
| shuttle | 49,097 | 9 | 3,511 (7.2%) |
| thyroid | 3,772 | 6 | 93 (2.5%) |

Table 2 shows the results of the competitive methods. We observe that the shallow kernel methods seem to perform slightly better on the rather small, low-dimensional benchmarks. Deep SAD proves competitive though and the small differences might be explained by the strong advantage we grant the shallow methods in the selection of their hyperparameters. We provide the complete table with the results from all methods in Appendix F

Table 2: Results on classic AD benchmark datasets in the setting with no pollution $\gamma_p = 0$ and a ratio of labeled anomalies of $\gamma_l = 0.01$ in the training set. We report avg. AUC with st. dev. computed over 10 seeds. A "$\star$" indicates a statistically significant ($\alpha = 0.05$) difference between 1st and 2nd.

| Dataset | OC-SVM Raw | OC-SVM Hybrid | Deep SVDD | SSAD Raw | SSAD Hybrid | Supervised Classifier | Deep SAD |
|---|---|---|---|---|---|---|---|
| arrhythmia | 84.5±3.9 | 76.7±6.2 | 74.6±9.0 | **86.7±4.0**$^\star$ | 78.3±5.1 | 39.2±9.5 | 75.9±8.7 |
| cardio | 98.5±0.3 | 82.8±9.3 | 84.8±3.6 | **98.8±0.3** | 86.3±5.8 | 83.2±9.6 | 95.0±1.6 |
| satellite | 95.1±0.2 | 68.6±4.8 | 79.8±4.1 | **96.2±0.3**$^\star$ | 86.9±2.8 | 87.2±2.1 | 91.5±1.1 |
| satimage-2 | 99.4±0.8 | 96.7±2.1 | 98.3±1.4 | **99.9±0.1** | 96.8±2.1 | **99.9±0.1** | **99.9±0.1** |
| shuttle | 99.4±0.9 | 94.1±9.5 | 86.3±7.5 | **99.6±0.5** | 97.7±1.0 | 95.1±8.0 | 98.4±0.9 |
| thyroid | 98.3±0.9 | 91.2±4.0 | 72.0±9.7 | 97.9±1.9 | 95.3±3.1 | 97.8±2.6 | **98.6±0.9** |

## C  OPTIMIZATION OF DEEP SAD

Our Deep SAD objective (7) is generally non-convex in the network weights $\mathcal{W}$ which usually is the case in deep learning. For a computationally efficient optimization, we rely on (mini-batch) SGD to optimize the network weights using backpropagation. For improved generalization, we add $L^2$ weight decay regularization with hyperparameter $\lambda > 0$ to the objective. Algorithm 1 summarizes the Deep SAD optimization routine.

---

**Algorithm 1** Optimization of Deep SAD

---

**Input:**
  Unlabeled data: $\boldsymbol{x}_1, \ldots, \boldsymbol{x}_n$
  Labeled data: $(\boldsymbol{x}_1', y_1'), \ldots, (\boldsymbol{x}_m', y_m')$
  Hyperparameters: $\eta, \lambda$
  SGD learning rate: $\varepsilon$
**Output:**
  Trained model: $\mathcal{W}^*$

  1: **Initialize:**
        Neural network weights: $\mathcal{W}$
        Hypersphere center: $\boldsymbol{c}$
  2: **for** each epoch **do**
  3:    **for** each mini-batch **do**
  4:       Draw mini-batch $\mathcal{B}$
  5:       $\mathcal{W} \leftarrow \mathcal{W} - \varepsilon \cdot \nabla_{\mathcal{W}} J(\mathcal{W}; \mathcal{B})$
  6:    **end for**
  7: **end for**

---

Using SGD allows Deep SAD to scale with large datasets as the computational complexity scales linearly in the number of training batches and computations in each batch can be parallelized (e.g., by training on GPUs). Moreover, Deep SAD has low memory complexity as a trained model is fully characterized by the final network parameters $\mathcal{W}^*$ and no data must be saved or referenced for prediction. Instead, the prediction only requires a forward pass on the network which usually is just a concatenation of simple functions. This enables fast predictions for Deep SAD.

**Initialization of the network weights** $\mathcal{W}$  We establish an autoencoder pre-training routine for initialization. That is, we first train an autoencoder that has an encoder with the same architecture as network $\phi$ on the reconstruction loss (mean squared error or cross-entropy). After training, we then initialize $\mathcal{W}$ with the converged parameters of the encoder. Note that this is in line with the Infomax principle (2) for unsupervised representation learning (Vincent et al., 2008).

**Initialization of the center** $\boldsymbol{c}$  After initializing the network weights $\mathcal{W}$, we fix the hypersphere center $\boldsymbol{c}$ as the mean of the network representations that we obtain from an initial forward pass on the data (excluding labeled anomalies). We found SGD convergence to be smoother and faster by fixing center $\boldsymbol{c}$ in the neighborhood of the initial data representations as also observed by Ruff et al. (2018). If sufficiently many labeled normal examples are available, using only those examples for a mean initialization would be another strategy to minimize possible distortions from polluted unlabeled training data. Adding center $\boldsymbol{c}$ as a free optimization variable would allow a trivial "hypersphere collapse" solution for the fully unlabeled setting, i.e. for unsupervised Deep SVDD.

**Preventing a hypersphere collapse**  A "hypersphere collapse" describes the trivial solution that neural network $\phi$ converges to the constant function $\phi \equiv \boldsymbol{c}$, i.e. the hypersphere collapses to a single point. Ruff et al. (2018) demonstrate theoretical network properties that prevent such a collapse which we adopt for Deep SAD. Most importantly, network $\phi$ must have no bias terms and no bounded activation functions. We refer to Ruff et al. (2018) for further details. If there are sufficiently many labeled anomalies available for training, however, hypersphere collapse is not a problem for Deep SAD due to the opposing labeled and unlabeled objectives.

# D  NETWORK ARCHITECTURES

We employ LeNet-type convolutional neural networks (CNNs) on MNIST, Fashion-MNIST, and CIFAR-10, where each convolutional module consists of a convolutional layer followed by leaky ReLU activations with leakiness $\alpha = 0.1$ and $(2 \times 2)$-max-pooling. On MNIST, we employ a CNN with two modules, $8 \times (5 \times 5)$-filters followed by $4 \times (5 \times 5)$-filters, and a final dense layer of 32 units. On Fashion-MNIST, we employ a CNN also with two modules, $16 \times (5 \times 5)$-filters and $32 \times (5 \times 5)$-filters, followed by two dense layers of 64 and 32 units respectively. On CIFAR-10, we employ a CNN with three modules, $32 \times (5 \times 5)$-filters, $64 \times (5 \times 5)$-filters, and $128 \times (5 \times 5)$-filters, followed by a final dense layer of 128 units.

On the classic AD benchmark datasets, we employ standard MLP feed-forward architectures. On arrhythmia, a 3-layer MLP with 128-64-32 units. On cardio, satellite, satimage-2, and shuttle a 3-layer MLP with 32-16-8 units. On thyroid a 3-layer MLP with 32-16-4 units.

For the (convolutional) autoencoders, we always employ the above architectures for the encoder networks and then construct the decoder networks symmetrically, where we replace max-pooling with simple upsampling and convolutions with deconvolutions.

# E  DETAILS ON COMPETING METHODS

**OC-SVM/SVDD**  The OC-SVM and SVDD are equivalent for the Gaussian/RBF kernel we employ. As mentioned in the main paper, we deliberately grant the OC-SVM/SVDD an unfair advantage by selecting its hyperparameters to maximize AUC on a subset (10%) of the test set to establish a strong baseline. To do this, we consider the RBF scale parameter $\gamma \in \{2^{-7}, 2^{-6}, \dots 2^2\}$ and select the best performing one. Moreover, we always repeat this over $\nu$-parameter $\nu \in \{0.01, 0.05, 0.1, 0.2, 0.5\}$ and then report the best final result.

**Isolation Forest (IF)**  We set the number of trees to $t = 100$ and the sub-sampling size to $\psi = 256$, as recommended in the original work (Liu et al., 2008).

**Kernel Density Estimator (KDE)**  We select the bandwidth $h$ of the Gaussian kernel from $h \in \{2^{0.5}, 2^1, \dots, 2^5\}$ via 5-fold cross-validation using the log-likelihood score following (Ruff et al., 2018).

**SSAD**  We also deliberately grant the state-of-the-art semi-supervised AD kernel method SSAD the unfair advantage of selecting its hyperparameters optimally to maximize AUC on a subset (10%) of the test set. To do this, we again select the scale parameter $\gamma$ of the RBF kernel we use from $\gamma \in \{2^{-7}, 2^{-6}, \dots 2^2\}$ and select the best performing one. Otherwise we set the hyperparameters as recommend by the original authors to $\kappa = 1$, $\kappa = 1$, $\eta_u = 1$, and $\eta_l = 1$ (Görnitz et al., 2013).

**(Convolutional) Autoencoder ((C)AE)**  To create the (convolutional) autoencoders, we symmetrically construct the decoders w.r.t. the architectures reported in Appenidx D, which make up the encoder parts of the autoencoders. Here, we replace max-pooling with simple upsampling and convolutions with deconvolutions. We train the autoencoders on the MSE reconstruction loss that also serves as the anomaly score.

**Hybrid Variants**  To establish hybrid methods, we apply the OC-SVM, IF, KDE, and SSAD as outlined above to the resulting bottleneck representations given by the respective converged autoencoders.

**Unsupervised Deep SVDD**  We consider both variants, Soft-Boundary Deep SVDD and One-Class Deep SVDD as unsupervised baselines and always report the better performance as the unsupervised result. For Soft-Boundary Deep SVDD, we optimally solve for the radius $R$ on every mini-batch and run experiments for $\nu \in \{0.01, 0.1\}$. We set the weight decay hyperparameter to $\lambda = 10^{-6}$. For Deep SVDD, we always remove all the bias terms from a network to prevent a hypersphere collapse as recommended by the authors in the original work (Ruff et al., 2018).

**Deep SAD**  We set $\lambda = 10^{-6}$ and equally weight the unlabeled and labeled examples by setting $\eta = 1$ if not reported otherwise.

**SS-DGM**  We consider both the M2 and M1+M2 model and always report the better performing result. Otherwise we follow the settings as recommended in the original work (Kingma et al., 2014).

Note that we use the latent *class probability estimate* (normal vs. anomalous) of semi-supervised DGM as a natural choice for the anomaly score, and *not* the reconstruction error as used for unsupervised autoencoding models such as the (convolutional) autoencoder we consider. Such deep semi-supervised models designed for classification as the downstream task have no notion of out-of-distribution and again implicitly make the cluster assumption (Zhu, 2005; Chapelle et al., 2009) we refer to. Thus, semi-supervised DGM also suffers from overfitting to previously seen anomalies at training similar to the supervised model which explains its bad AD performance.

**Supervised Deep Binary Classifier** To interpret AD as a binary classification problem, we rely on the typical assumption that most of the unlabeled training data is normal by assigning $y = +1$ to all unlabeled examples. Already labeled normal examples and labeled anomalies retain their assigned labels of $\tilde{y} = +1$ and $\tilde{y} = -1$ respectively. We train the supervised classifier on the binary cross-entropy loss. Note that in scenario (i), in particular, the supervised classifier has perfect, unpolluted label information but still fails to generalize as there are novel anomaly classes at testing.

**SGD Optimization Details for Deep Methods** We use the Adam optimizer with recommended default hyperparameters (Kingma & Ba, 2015) and apply Batch Normalization (Ioffe & Szegedy, 2015) in SGD optimization. For all deep approaches and on all datasets, we employ a two-phase ("searching" and "fine-tuning") learning rate schedule. In the searching phase we first train with a learning rate $\varepsilon = 10^{-4}$ for 50 epochs. In the fine-tuning phase we train with $\varepsilon = 10^{-5}$ for another 100 epochs. We always use a batch size of 200. For the autoencoder, SS-DGM, and the supervised classifier, we initialize the network with uniform Glorot weights (Glorot & Bengio, 2010). For Deep SVDD and Deep SAD, we establish an unsupervised pre-training routine via autoencoder as explained in Appendix C, where we set the network $\phi$ to be the encoder of the autoencoder that we train beforehand.

## F    Complete Tables of Experimental Results

The following Tables 3–6 list the complete experimental results of all the methods in all our experiments.

Table 3: Complete results of experimental scenario (i), where we increase the ratio of labeled anomalies $\gamma_l$ in the training set. We report the avg. AUC with st. dev. computed over 90 experiments at various ratios $\gamma_l$.

| Data | $\gamma_l$ | OC-SVM Raw | OC-SVM Hybrid | IF Raw | IF Hybrid | KDE Raw | KDE Hybrid | CAE | Deep SVDD | SSAD Raw | SSAD Hybrid | SS-DGM | Deep SAD | Supervised Classifier |
|---|---|---|---|---|---|---|---|---|---|---|---|---|---|---|
| MNIST | .00 | 96.0±2.9 | 96.3±2.5 | 85.4±8.7 | 90.5±5.3 | 95.0±3.3 | 87.8±5.6 | 92.9±5.7 | 92.8±4.9 | 96.0±2.9 | 96.3±2.5 | | 92.8±4.9 | |
| | .01 | | | | | | | | | 96.6±2.4 | 96.8±2.3 | 89.9±9.2 | 96.4±2.7 | 92.8±5.5 |
| | .05 | | | | | | | | | 93.3±3.6 | 97.4±2.0 | 92.2±5.6 | 96.7±2.4 | 94.5±4.6 |
| | .10 | | | | | | | | | 90.7±4.4 | 97.6±1.7 | 91.6±5.5 | 96.9±2.3 | 95.0±4.7 |
| | .20 | | | | | | | | | 87.2±5.6 | 97.8±1.5 | 91.2±5.6 | 96.9±2.4 | 95.6±4.4 |
| F-MNIST | .00 | 92.8±4.7 | 91.2±4.7 | 91.6±5.5 | 82.5±8.1 | 92.0±4.9 | 69.7±14.4 | 90.2±5.8 | 89.2±6.2 | 92.8±4.7 | 91.2±4.7 | | 89.2±6.2 | |
| | .01 | | | | | | | | | 92.1±5.0 | 89.4±6.0 | 65.1±16.3 | 90.0±6.4 | 74.4±13.6 |
| | .05 | | | | | | | | | 88.3±6.2 | 90.5±5.9 | 71.4±12.7 | 90.5±6.5 | 76.8±13.2 |
| | .10 | | | | | | | | | 85.5±7.1 | 91.0±5.6 | 72.9±12.2 | 91.3±6.0 | 79.0±12.3 |
| | .20 | | | | | | | | | 82.0±8.0 | 89.7±6.6 | 74.7±13.5 | 91.0±5.5 | 81.4±12.0 |
| CIFAR-10 | .00 | 62.0±10.6 | 63.8±9.0 | 60.0±10.0 | 59.9±6.7 | 59.9±11.7 | 56.1±10.2 | 56.2±13.2 | 60.9±9.4 | 62.0±10.6 | 63.8±9.0 | | 60.9±9.4 | |
| | .01 | | | | | | | | | 73.0±8.0 | 70.5±8.3 | 49.7±1.7 | 72.6±7.4 | 55.6±5.0 |
| | .05 | | | | | | | | | 71.5±8.1 | 73.3±8.4 | 50.8±4.7 | 77.9±7.2 | 63.5±8.0 |
| | .10 | | | | | | | | | 70.1±8.1 | 74.0±8.1 | 52.0±5.5 | 79.8±7.1 | 67.7±9.6 |
| | .20 | | | | | | | | | 67.4±8.8 | 74.5±8.0 | 53.2±6.7 | 81.9±7.0 | 80.5±5.9 |

Table 4: Complete results of experimental scenario (ii), where we pollute the unlabeled part of the training set with (unknown) anomalies. We report the avg. AUC with st. dev. computed over 90 experiments at various ratios $\gamma_p$.

| Data | $\gamma_p$ | OC-SVM Raw | OC-SVM Hybrid | IF Raw | IF Hybrid | KDE Raw | KDE Hybrid | CAE | Deep SVDD | SSAD Raw | SSAD Hybrid | SS-DGM | Deep SAD | Supervised Classifier |
|---|---|---|---|---|---|---|---|---|---|---|---|---|---|---|
| MNIST | .00 | 96.0±2.9 | 96.3±2.5 | 85.4±8.7 | 90.5±5.3 | 95.0±3.3 | 87.8±5.6 | 92.9±5.7 | 92.8±4.9 | 97.9±1.8 | 97.4±2.0 | 92.2±5.6 | 96.7±2.4 | 94.5±4.6 |
| | .01 | 94.3±3.9 | 95.6±2.5 | 85.2±8.8 | 90.6±5.0 | 91.2±4.9 | 87.9±5.3 | 91.3±6.1 | 92.1±5.1 | 96.6±2.4 | 95.2±2.3 | 92.0±6.0 | 95.5±3.3 | 91.5±5.9 |
| | .05 | 91.4±5.2 | 93.8±3.9 | 83.9±9.2 | 89.7±6.0 | 85.5±7.1 | 87.3±7.0 | 87.2±7.1 | 89.4±5.8 | 93.4±3.4 | 89.5±3.9 | 91.0±6.9 | 93.5±4.1 | 86.7±7.4 |
| | .10 | 88.8±6.0 | 91.4±5.1 | 82.3±9.5 | 88.2±6.5 | 82.1±8.5 | 85.9±6.6 | 83.7±8.4 | 86.5±6.8 | 90.7±4.4 | 86.0±4.6 | 89.7±7.5 | 91.2±4.9 | 83.6±8.2 |
| | .20 | 84.1±7.6 | 85.9±7.6 | 78.7±10.5 | 85.3±7.9 | 77.4±10.9 | 82.6±8.6 | 78.6±10.3 | 81.5±8.4 | 87.4±5.6 | 82.1±5.4 | 87.4±8.6 | 86.6±6.6 | 79.7±9.4 |
| F-MNIST | .00 | 92.8±4.7 | 91.2±4.7 | 91.6±5.5 | 82.5±8.1 | 92.0±4.9 | 69.7±14.4 | 90.2±5.8 | 89.2±6.2 | 94.0±4.4 | 90.5±5.9 | 71.4±12.7 | 90.5±6.5 | 76.8±13.2 |
| | .01 | 91.7±5.0 | 91.5±4.6 | 91.5±5.5 | 84.9±7.2 | 89.4±6.3 | 73.9±12.4 | 87.1±7.3 | 86.3±6.3 | 92.2±4.9 | 87.8±6.1 | 71.2±14.3 | 87.2±7.1 | 67.3±8.1 |
| | .05 | 90.7±5.5 | 90.7±4.9 | 90.9±5.9 | 85.5±7.2 | 85.2±9.1 | 75.4±12.9 | 81.6±9.6 | 80.6±7.1 | 88.3±6.2 | 82.7±7.8 | 71.9±14.3 | 81.5±8.5 | 59.8±4.6 |
| | .10 | 89.5±6.1 | 89.3±6.2 | 90.2±6.3 | 85.5±7.7 | 81.8±11.2 | 77.8±12.0 | 77.4±11.1 | 76.2±7.3 | 85.6±7.0 | 79.8±9.0 | 72.5±15.5 | 78.2±9.1 | 56.7±4.1 |
| | .20 | 86.3±7.7 | 88.1±6.9 | 88.4±7.6 | 86.3±7.4 | 77.4±13.6 | 82.1±9.8 | 72.5±12.6 | 69.3±6.3 | 81.9±8.1 | 74.3±10.6 | 70.8±16.0 | 74.8±9.4 | 53.9±2.9 |
| CIFAR-10 | .00 | 62.0±10.6 | 63.8±9.0 | 60.0±10.0 | 59.9±6.7 | 59.9±11.7 | 56.1±10.2 | 56.2±13.2 | 60.9±9.4 | 73.8±7.6 | 73.3±8.4 | 50.8±4.7 | 77.9±7.2 | 63.5±8.0 |
| | .01 | 61.9±10.6 | 63.8±9.3 | 59.9±10.1 | 59.9±6.7 | 59.2±12.3 | 56.3±10.4 | 56.2±13.1 | 60.5±9.4 | 73.0±8.0 | 72.8±8.1 | 51.1±4.7 | 76.5±7.2 | 62.9±7.3 |
| | .05 | 61.4±10.7 | 62.6±9.2 | 59.6±10.1 | 59.6±6.4 | 58.1±12.9 | 55.6±10.5 | 55.7±13.3 | 59.6±9.8 | 71.5±8.2 | 71.0±8.4 | 50.1±2.9 | 74.0±6.9 | 62.2±8.2 |
| | .10 | 60.8±10.7 | 62.9±8.2 | 58.8±10.1 | 59.1±6.6 | 57.3±13.5 | 54.9±11.1 | 55.4±13.3 | 58.6±10.0 | 69.8±8.4 | 69.3±8.5 | 50.5±3.6 | 71.8±7.0 | 60.6±8.3 |
| | .20 | 60.3±10.3 | 61.9±8.1 | 57.9±10.1 | 58.3±6.2 | 56.2±13.9 | 54.2±11.1 | 54.6±13.3 | 57.0±10.6 | 67.8±8.6 | 67.9±8.1 | 50.1±1.7 | 68.5±7.1 | 58.5±6.7 |

Table 5: Complete results of experimental scenario (iii), where we increase the number of anomaly classes $k_l$ included in the labeled training data. We report the avg. AUC with st. dev. computed over 100 experiments at various numbers $k_l$.

| Data | $k_l$ | OC-SVM Raw | OC-SVM Hybrid | IF Raw | IF Hybrid | KDE Raw | KDE Hybrid | CAE | Deep SVDD | SSAD Raw | SSAD Hybrid | SS-DGM | Deep SAD | Supervised Classifier |
|---|---|---|---|---|---|---|---|---|---|---|---|---|---|---|
| MNIST | 0 | 88.8±6.0 | 91.4±5.1 | 82.3±9.5 | 88.2±6.5 | 82.1±8.5 | 85.9±6.6 | 83.7±8.4 | 86.5±6.8 | 88.8±6.0 | 91.4±5.1 | | 86.5±6.8 | |
| | 1 | | | | | | | | | 90.7±4.4 | 86.0±4.6 | 89.7±7.5 | 91.2±4.9 | 83.6±8.2 |
| | 2 | | | | | | | | | 92.5±3.6 | 87.7±3.8 | 92.8±5.3 | 92.0±3.6 | 90.3±4.6 |
| | 3 | | | | | | | | | 93.9±3.3 | 89.8±3.3 | 94.9±4.2 | 94.7±2.8 | 93.9±2.8 |
| | 5 | | | | | | | | | 95.5±2.5 | 91.9±3.0 | 96.7±2.3 | 97.3±1.8 | 96.9±1.7 |
| F-MNIST | 0 | 89.5±6.1 | 89.3±6.2 | 90.2±6.3 | 85.5±7.7 | 81.8±11.2 | 77.8±12.0 | 77.4±11.1 | 76.2±7.3 | 89.5±6.1 | 89.3±6.2 | | 76.2±7.3 | |
| | 1 | | | | | | | | | 85.6±7.0 | 79.8±9.0 | 72.5±15.5 | 78.2±9.1 | 56.7±4.1 |
| | 2 | | | | | | | | | 87.8±6.1 | 80.1±10.5 | 74.3±15.4 | 80.5±8.2 | 62.3±2.9 |
| | 3 | | | | | | | | | 89.4±5.5 | 83.8±9.4 | 77.5±14.7 | 83.9±7.4 | 67.3±3.0 |
| | 5 | | | | | | | | | 91.2±4.8 | 86.8±7.7 | 79.9±13.8 | 87.3±6.4 | 75.3±2.7 |
| CIFAR-10 | 0 | 60.8±10.7 | 62.9±8.2 | 58.8±10.1 | 59.1±6.6 | 57.3±13.5 | 54.9±11.1 | 55.4±13.3 | 58.6±10.0 | 60.8±10.7 | 62.9±8.2 | | 58.6±10.0 | |
| | 1 | | | | | | | | | 69.8±8.4 | 69.3±8.5 | 50.5±3.6 | 71.8±7.0 | 60.6±8.3 |
| | 2 | | | | | | | | | 73.0±7.1 | 72.3±7.5 | 50.3±2.4 | 75.2±6.4 | 61.0±6.6 |
| | 3 | | | | | | | | | 73.8±6.6 | 73.3±7.0 | 50.0±0.7 | 77.5±5.9 | 62.7±6.8 |
| | 5 | | | | | | | | | 75.1±5.5 | 74.2±6.5 | 50.0±1.0 | 80.4±4.6 | 60.9±4.6 |

Table 6: Complete results on classic AD benchmark datasets in the setting with no pollution $\gamma_p = 0$ and a ratio of labeled anomalies of $\gamma_l = 0.01$ in the training set. We report the avg. AUC with st. dev. computed over 10 seeds.

| Data | OC-SVM Raw | OC-SVM Hybrid | CAE | Deep SVDD | SSAD Raw | SSAD Hybrid | SS-DGM | Deep SAD | Supervised Classifier |
|------|-----------|---------------|-----|-----------|----------|-------------|--------|----------|----------------------|
| arrhythmia | 84.5±3.9 | 76.7±6.2 | 74.0±7.5 | 74.6±9.0 | 86.7±4.0 | 78.3±5.1 | 50.3±9.8 | 75.9±8.7 | 39.2±9.5 |
| cardio | 98.5±0.3 | 82.8±9.3 | 94.3±2.0 | 84.8±3.6 | 98.8±0.3 | 86.3±5.8 | 66.2±14.3 | 95.0±1.6 | 83.2±9.6 |
| satellite | 95.1±0.2 | 68.6±4.8 | 80.0±1.7 | 79.8±4.1 | 96.2±0.3 | 86.9±2.8 | 57.4±6.4 | 91.5±1.1 | 87.2±2.1 |
| satimage-2 | 99.4±0.8 | 96.7±2.1 | 99.9±0.0 | 98.3±1.4 | 99.9±0.1 | 96.8±2.1 | 99.2±0.6 | 99.9±0.1 | 99.9±0.1 |
| shuttle | 99.4±0.9 | 94.1±9.5 | 98.2±1.2 | 86.3±7.5 | 99.6±0.5 | 97.7±1.0 | 97.9±0.3 | 98.4±0.9 | 95.1±8.0 |
| thyroid | 98.3±0.9 | 91.2±4.0 | 75.2±10.2 | 72.0±9.7 | 97.9±1.9 | 95.3±3.1 | 72.7±12.0 | 98.6±0.9 | 97.8±2.6 |

