# OpenReview forum: "Deep Semi-Supervised Anomaly Detection"
_ICLR.cc/2020/Conference — Accept (Poster)_

### Official Review · AnonReviewer3 · 2019-10-23
**Official Blind Review #854**

**Rating:** 6

**Review:**


Summary of the work
- The work proposes a new method two find anomaly (out of distribution) data when some labeled anomalies are given.
- The authors apply information theory-derived loss based on that the normal (in distribution) data usually have lower entropy compared to that of the abnormal data.
- The paper conducts extensive experiments on MNIST, Fashion-MNIST, and CIFAR 10, with varying the number of labeled anomlies.

I think the paper is well written and the experiment seems to support the authors argument. Unfortunately, this field is not overlapped to my research field, and it is hard for me to judge this paper.

**Experience Assessment:**

I do not know much about this area.

**Review Assessment: Checking Correctness Of Derivations And Theory:**

I assessed the sensibility of the derivations and theory.

**Review Assessment: Checking Correctness Of Experiments:**

I assessed the sensibility of the experiments.

**Review Assessment: Thoroughness In Paper Reading:**

I read the paper at least twice and used my best judgement in assessing the paper.

---

### Official Review · AnonReviewer2 · 2019-10-24
**Official Blind Review #2**

**Rating:** 6

**Review:**

[Summary]
The paper proposes an abnormal detection (AD) framework under general settings where 1) unlabeled data, 2) labeled positive (normal) data, and 3) labeled negative (abnormal) data are available (with the last two optional), denoted as semi-supervised AD. Starting from the assumption that abnormal data are sampled from background unpredicted distribution, rather than “cluster” assumption, it is argued that conventional discriminative formulation is not applicable. Motivated by the recent deep AD methods (e.g., deep SVDD), the paper proposes to approach semi-supervised AD from the information theoretic perspective where 1) mutual information between raw data and learnt representation should be maximized (infomax principle), 2) entropy of labeled positive data should be minimized (“compactness” constraint), and 3) enrtropy of labeled negative data should be maximized to reflect the uncertainty assumption of anomaly. The solution is implemented by the encoder of a pre-trained autoencoder that is further fine tuned to enforce entropy assumption on all types of training data. Extensive experiments on benchmarks suggests promising results on the proposed framework versus other state-of-the-arts.

[Comments]
The paper is well written and easy to follow (the presentation is especially pleasant to read). The problem is well defined and of interest to the community under fairly general and practical conditions. Despite the fact that the implementation is only marginally tweaked from previous work (deep SVDD), the theoretical motivation, nevertheless, is sound and well justified, and the empirical evaluation is extensive to reveal the behaviors of the proposed method. It would be better if complexity analysis can also be provided for all concerning methods. Overall, the value of the paper is worth circulation in the community.

[Area to improve]
The manuscript could be further improved by exploring the training process more. In the current format, the solution follows the strategy of deep SVDD that learns the model in two separate stages: pre-training the autoencoder, and then fitting the encoder to enforce compactness and entropy minimization/maximization. What if these are implemented in an end-to-end fashion? Will this help to achieve a better result?


**Experience Assessment:**

I have published one or two papers in this area.

**Review Assessment: Checking Correctness Of Derivations And Theory:**

I carefully checked the derivations and theory.

**Review Assessment: Checking Correctness Of Experiments:**

I assessed the sensibility of the experiments.

**Review Assessment: Thoroughness In Paper Reading:**

I read the paper at least twice and used my best judgement in assessing the paper.

---

> ### Author Response · Authors · 2019-11-14
> **Author Response to Review #2**
>
> We have performed experiments without pre-training as well as incorporating end-to-end autoencoder training. Our method consistently performs worse without pre-training. With an end-to-end autoencoder we achieve performance approximately as good as presented in this paper, provided one uses a training regimen which first emphasizes and then de-emphasizes the autoencoder loss, which is essentially pre-training. We will include these points in our final draft.

---

### Official Review · AnonReviewer1 · 2019-10-24
**Official Blind Review #1**

**Rating:** 6

**Review:**

The authors propose in this paper a variant of Deep SVDD which brings semi-supervision to this model. The paper is well written and contains a thorough experimental evaluation (even disregarding the supplementary material). As far as I know, the proposed method is new and improve anomaly detection. The modification of Deep SVDD are in a way minimalistic, but they do the job.

The only negative aspects, in my opinion, is the "information-theoretic view" which is close to hand waving. The authors are indeed 1) disregarding the regularization term 2) considering an upper bound of the entropy 3) pretending the results on the pre-trained NN hold after post training. Putting everything together, I do not see how this reasoning could accepted. In fact, its extension to Deep SVDD is even more problematic as the discussion in the paper contradicts the reasoning. The authors emphasize the fact that anomalies should not fulfill the clustering assumption (which is indeed an important remark). But then the distribution of phi(x,W) cannot be approximated by a Gaussian for anomalies and thus the bound on the entropy is not valid.

I strongly recommend to remove this part of the paper and to derive Deep SAD from Deep SVDD from heuristics consideration (which is fine!). This will provide an opportunity to remove the cute sentence "We are happy to now introduce Deep SAD".

**Experience Assessment:**

I have read many papers in this area.

**Review Assessment: Checking Correctness Of Derivations And Theory:**

I assessed the sensibility of the derivations and theory.

**Review Assessment: Checking Correctness Of Experiments:**

I assessed the sensibility of the experiments.

**Review Assessment: Thoroughness In Paper Reading:**

I read the paper at least twice and used my best judgement in assessing the paper.

---

> ### Author Response · Authors · 2019-11-14
> **Author Response to Review #1**
>
> We understand your concerns with the "information-theoretic view," nonetheless we think the framework should remain and is of interest to the ML community for a few reasons:
>
> - Previous theoretical work on anomaly detection [1,2] often assumed that the anomalous distribution is unconcentrated (analogously high-entropy or not clustered) and it is reasonable that this intuition would extend to deep anomaly detection. Our framework introduces and explores the idea of using a training objective to incorporate this assumption. This presents an avenue for future work. For example, in our paper we use high-variance as a proxy for distribution not being clustered. Can we find a better loss to enforce anomalies to be unclustered? Further future work could study the detection performance under the information-theoretic rate-distortion curve [3] or derive novel methods from using other non-parametric estimators for mutual information (e.g. [4]) based on our framework.
>
> - Other recent work on deep anomaly detection utilizing self-supervised classification have incorporated the use of anomalies during training in a similar way, albeit without theoretical justification [5]. In these methods normal samples are trained to minimize a classification loss. On the other hand anomalous samples are trained so that the softmax output distribution has high entropy, not for misclassification. This results in a network where the softmax output from normal samples are concentrated around at the corners of the probability simplex, and anomalous samples are diffusely spread around the center. Our information-theoretic framework offers a potential explanation for why such an objective is natural and connects it to our method.
>
> - The Gaussianity assumption for $Z$ used in Eq. (6) is merely a choice of convenience in order to obtain a simple derivation, but is not necessary to derive the claim that the entropy is minimized by minimizing the empirical variance. To see this, note that the upper bound in Eq. (5) is a log determinant of covariance $\Sigma$, which is a sum of the log eigenvalues of $\Sigma$. These eigenvalues are variances obtained by projecting the data on certain orthonormal basis vectors $r_e$. By Cauchy-Schwarz inequality, each of these projected variances $E[\|  (Z \cdot r_e-E[Z \cdot r_e]) \|^2]$ are upper bounded by the scalar variance $E[\|  (Z-E[Z]) \|^2]$ since $\|r_e\|=1$.
> We will add a complete derivation of this claim to the appendix.
>
> - Reviewer 2 expressed interest in our framework, so we are reluctant to remove it completely.
>
> In a final version of the paper we will elaborate on the connection to previous work, with a focus on future work. We will also de-emphasize this framework as a way to derive our method, treating it instead as a possible interpretation for why our loss is somewhat "natural," e.g. why it makes sense to concentrate the nominal samples and let the anomalies be diffuse. We will also remove our "cute sentence."
>
>
> [1] Steinwart, I., Hush, D., and Scovel, C. A Classification Framework for Anomaly Detection. Journal of Machine Learning Research, 6(Feb):211–232, 2005.
> [2] Schölkopf, B. et al. Estimating the Support of a High-Dimensional Distribution. Neural computation, 13(7):1443–1471, 2001.
> [3] Alemi, A. et al. Fixing a broken ELBO. In ICML, volume 80, pages 159–168, 2018.
> [4] Hjelm, R. D. et al. Learning Deep Representations by Mutual Information Estimation and Maximization. In ICLR, 2019.
> [5] Hendrycks, D. Mazeika, M., and Dietterich, T. G. Deep Anomaly Detection with Outlier Exposure. In ICLR, 2019.

---

### Public Comment · ~Guansong_Pang1 · 2019-11-06
**Nice work but closely related references are missing**

The problem of using a few labeled data , especially limited labeled anomalies, to enhance anomaly detection performance is very important, as such settings are common in real-world applications and we can definitely obtain much better performance than fully unsupervised methods when the labeled data leveraged properly. The authors extend the deep svdd method and provide comprehensive and interesting empirical results. Some main concerns I have are as follows:
1. some closely related reference s are missing, especially studies on exploiting a small number of labeled anomalies to achieve deep anomaly detection, such as
"Deep Anomaly Detection with Deviation Networks". In Proceedings of the 25th ACM SIGKDD International Conference on Knowledge Discovery & Data Mining (pp. 353-362), 2019. ACM. Actually, labeled normal data does not contribute much to the improved performance, as a large number of normal samples exists in the unlabeled data and can be well captured by the deep svdd term.  Thus, labeled anomaly data plays the critical role here.  The above KDD paper extensively examined this point, including the impact of increasing number of anomalies and increasing anomaly contamination rate.  I suggest the authors to have some discussion of the key differences between their work and those closely relevant work.

2. Also, currently the competing method list  does not contain semi-supervised anomaly detection methods, which seems to be incomplete.  Comparing to the method in the KDD paper may help well address this problem.

---

> ### Author Response · Authors · 2019-11-15
> **Related Work**
>
> Thank you very much for your comment. We will add your recent paper to our related work. Note that we do compare to semi-supervised anomaly detection methods such as (hybrid) SSAD in the paper.

---

### Author Response · Authors · 2019-11-14
**General Author Response**

We kindly thank the reviewers for taking the time to review our paper and are pleased that our work has been well received overall. The comments of reviewers 1 and 2 were particularly helpful. We found reviews reasonable and there are no particular points we feel need to be rebutted. Indeed the reviewers questions and concerns align with our own and we will incorporate their points into a final draft. We provide our answers to specific questions in individual comments.

---

### Decision · Program_Chairs · 2019-12-19

**Decision:**

Accept (Poster)

**Comment:**

Issues raised by the reviewers have been addressed by the authors, and thus I suggest the acceptance of this paper.